# Exploring the Comprehensive Kozak Sequence Landscape for AAV Production in Sf9 System

**DOI:** 10.3390/v15101983

**Published:** 2023-09-23

**Authors:** Oleksandr Kondratov, Sergei Zolotukhin

**Affiliations:** Division of Cellular and Molecular Therapy, Department of Pediatrics, College of Medicine, University of Florida, Gainesville, FL 32610, USA; szlt@ufl.edu

**Keywords:** rAAV, Kozak, Sf9, packaging, transduction, directed evolution

## Abstract

The widespread successful use of recombinant Adeno-associated virus (rAAV) in gene therapy has driven the demand for scale-up manufacturing methods of vectors with optimized yield and transduction efficiency. The Baculovirus/Sf9 system is a promising platform for high yield production; however, a major drawback to using an invertebrate cell line compared to a mammalian system is a generally altered AAV capsid stoichiometry resulting in lower biological potency. Here, we introduce a term of the structural and biological “fitness” of an AAV capsid as a function of two interdependent parameters: (1) packaging efficiency (yield), and (2) transduction efficiency (infectivity). Both parameters are critically dependent on AAV capsid structural proteins VP1/2/3 stoichiometry. To identify an optimal AAV capsid composition, we developed a novel Directed Evolution (DE) protocol for assessing the structural and biological fitness of Sf9-manufactured rAAV for any given serotype. The approach involves the packaging of a combinatorial capsid library in insect Sf9 cells, followed by a library screening for high infectivity in human Cre–recombinase-expressing C12 cells. One single DE selection round, complemented by Next-Generation Sequencing (NGS) and guided by in silico analysis, identifies a small subset of VP1 translation initiation sites (known as Kozak sequence) encoding “fit” AAV capsids characterized by a high production yield and superior transduction efficiencies.

## 1. Introduction

Recombinant Adeno-Associated Virus demonstrates promising potential as a gene therapy delivery vehicle because of its low toxicity and immunogenicity, prolonged expression of the delivered genetic material, and the possibility of generating cell-type-specific rAAV targeting mutants [1]. There are multiple ongoing preclinical/clinical trials of rAAV gene therapy and seven of them are approved by FDA/EMA: Glybera, Luxturna, Zolgenesma, Hemgenix, Roctavian, Upstaza and Elevidys [2]. The encouraging results of pre-clinical trials sparked demand for the large-scale production of high-quality AAV vectors.

Originally, AAV production was established in HEK293 cells using a triple-transfection system that relies on genes of interest flanked by AAV inverted terminal repeats (ITRs), Rep/Cap, and Helper vectors [3]. Due to the fact that the production of rAAV in HEK293 grown in 2D format is labor-intensive and not scalable, several suspension cell-culture-based mammalian, insect, and yeast-based systems have been developed [4,5,6].

The Sf9 (*Spodoptera frugiperda*) is an insect cell line adapted for propagation both on plastic and in 3D suspension culture, making it extremely useful for rAAV production [7]. The original version of the Sf9 production system used three separate Baculovirus Expression Vectors (BEVs) to deliver ITR-cassette (Bac-ITR), and to express Rep and Cap proteins (BacRep and BacCap) [5,8]. In the Sf9 system, the baculoviral infection, in and of itself, provides all necessary helper functions for rAAV replication and assembly. Further improvements to the Sf9-Bac system were introduced, reducing the number of helpers to just two baculoviruses [9]. In addition, alternative start codons or integrated insect-specific introns to express the proper Cap protein ratio (VP1:VP2:VP3) were applied by several research groups for the optimization of VP ratio/infectivity [10]. These techniques demonstrated promising results, but only for certain AAV serotypes [11,12,13,14]. In order to overcome the problems of BEV stability/maintenance, and to improve AAV biological potency, we developed a novel approach that relies on only a single BEV [11,12] while utilizing a wide range of Kozak sequences for the precise tuning of capsid protein ratio (OneBac3.0) [15,16]. As described, the latter design is potentially applicable to all AAV serotypes featuring tunable parameters of production and infectivity for assembled AAV.

OneBac3.0 platform utilizes an attenuated Kozak sequence upstream of the canonical ATG initiation codon using a leaky ribosomal scanning to achieve the serotype-specific modulation of rAAV capsid proteins’ stoichiometry. Recently, we showed that the system with optimized Kozak generates rAAV5 with a six-fold reduction in collaterally packaged contaminating DNA compared to the HEK293 system. An analysis of transduction efficiency showed the equal (AAV9) or even higher biological potency (AAV5) of rAAV manufactured in the OneBac3.0 system compared with the HEK293 system. All these data indicate that it is an optimal platform for the large-scale production of clinical-grade recombinant rAAV vectors of any serotype.

To identify an optimal Kozak sequence for rAAV5 and 9, we used the data from exhaustive Kozak screenings in mammalian cells [17] to narrow down a set of potentially suitable Kozak variants. The routine screening of hundreds of mammalian Kozak sequences, however, is an extremely laborious process and cannot completely rely on the mammalian sequences dataset since the consensus translation initiation site (TIS) appears to be different in invertebrates [18]. To overcome these barriers, we designed a highly effective screening approach of 65,536 Kozak sequences for the insect-cell-based production of rAAV optimized for high yield and, simultaneously, high infectivity. Our approach utilizes one single round of the Directed Evolution (DE) facilitated by orthogonal selection in two cell lines, insect Sf9 and mammalian HeLa C12, Next-Generation Sequencing (NGS) analysis, and assisted by the Fitness Guided Directed Evolution (FGDE). Using this approach allowed us to narrow down the most optimized Kozak variants, combining two interrelated attributes that are critically important for the therapeutic application of AAV vectors: structural fitness and biological potency.

## 2. Materials and Methods

### 2.1. Construction of the Libraries and a Stable Cell Line

For the assembly of the pLox-Cap-KozakLib-EGFP (Appendix A) and Cre-expressing vectors, we used the respective genetic elements from the Addgene repository (plasmids #105840, #59701) [19]. In order to synthesize 65,536 Kozak variants, we used degenerate oligos with theoretically equal distribution of four possible nucleotides in eight positions: NNNNNNNN − 4^8^ = 65,536 combinations. We utilized the Lox66/Lox71 pair, as published earlier [20], because this Lox combination demonstrates a low rate of reverse inversion reaction. To establish tet-controlled Cre expression in C12 cell lines, we cloned the Cre recombinase-coding sequence into vector (plasmid #105840) after the excision of LoxP elements. To generate a Cre-expressing stable pooled cell line, HeLa C12 cells were transfected with the pTetCre-mCherry vector and selected for G418 antibiotic resistance for 3 weeks.

### 2.2. The Kozak-Cap rAAV Vector Assembly

To generate an AAV library with pLox-Cap-KozakLib-EGFP vector, we transfected 70% confluent Sf9 cells in SFM medium using Cellfectin II reagent (Gibco, Carlsbad, CA, USA) with PLUS™ Reagent (Invitrogen, Carlsbad, California, USA). For the transfection, we used the ratio of 500 plasmid copies per cell per 175 cm flask. This plasmid ratio was selected to avoid potential cross-packaging, which affects phenotype/genotype linkage [21]. One day post transfection, Sf9 cells were infected with BEV-BacRep (passage 3) with a multiplicity of infection (MOI) of 3. Seventy-two hours after BacRep infection, Sf9 cells were collected and processed for rAAV purification by iodixanol gradient with Benzonase treatment.

### 2.3. FGDE vs. Standard Approaches

Initially, we performed Lox–Cap–Kozak-directed evolution according to the standard scheme with the NNNNNN ATG NN combinatorial library (from here on referred to as (N)6ATGNN, whereby ATG is a canonical initiation codon) relying only on Cre-inverted DNA enrichment for 2 cycles. A day before transduction by the AAV-(N)6ATGNN library, we activated Cre expression in the HeLa C12Cre cell line by supplementing the medium with Dox antibiotic. The next day, we transduced 50% confluent HeLa C12Cre cells with AAV**-**(N)6ATGNN library. Twenty-four hours after AAV transduction, we co-transduced HeLa C12Cre cells with human adenovirus-5 (Ad5), MOI of 5. This step was necessary for the activation of dsDNA synthesis for the delivered AAV genomes. In the case of the standard DE protocol, two days after Ad5 co-transduction, we collected HeLa C12Cre cells and isolated total DNA using the EZNA kit. Total DNA was used for Cre-Lox inverted PCR with primers matching inverted sequences between Lox elements. In this way, Kozak sequences of successfully transduced variants were PCR-amplified and re-cloned into the pKozakCapLoxGFP vector for the second round of library reconstitution. The process was repeated one more time for a total of two rounds of DE.

For the FGDE protocol, we designed an alternative CapKozak library, NNNNNNNN ATG GC [(N)8ATGGC]. Modifying the selection protocol described above, HeLa C12Cre cells (Figure 1a) were transduced with AAV**-**(N)8ATGGC library (Figure 1b) and co-infected with Ad5 (MOI of 5) (Figure 1c). Forty-eight hours later, HeLa C12Cre cells were subjected to FACS sorting (Sony SH800) to enrich the GFP-positive fraction (Figure 1d,e). At this step, we isolated ~1–2 M of GFP^+^ cells covering the theoretical Cap–Kozak library complexity (65,536 variants) more than 10 times. Total DNA was isolated from FACS-enriched GFP+ HeLa C12Cre cells, amplified by Cre-inverted PCR (Figure 1f) and barcoded for subsequent Illumina sequencing. For NGS sequencing, we used NextSeq500 platform with 75 cycle settings (Figure 1g). To carry out Cre-inverted PCR, we used invFor/wtRev-Polh–Rev primers, which had Rev–Rev orientation before Cre-mediated inversion and For–Rev orientation in Cre-inverted DNA (Appendix A (pink). Subsequent PCR barcoding was performed by 8 cycles of amplification of gel-purified Cre-inverted PCR product with internal NGS-For/Rev (Appendix A (grey)) primers. All stages of amplification were carried out with Q5 polymerase. In order to comprehensively characterize Kozak variant distributions, we used ultra-deep coverage NGS (>300×) at the most diverse stage - plasmid library. Therefore, we dubbed this quantitative NGS (qNGS). Using the FGDE approach, one cycle of enrichment was sufficient to select a subset of optimal assemblers/transducers.

### 2.4. Bioinformatics Analysis

For the debarcoding of NGS raw data and calculation of Kozak variant distributions, we used a previously published python script [22] with all possible variants for eight ambiguous nucleotides (4^8^ = 65,536), which were substituted with an internal barcode library file (Figure 1h). For heatmap visualization, we used the principle of fractal representation of ambiguous DNA sequences with resolution 4 × 4 for both (N)6ATGNN and (N)8ATGGC libraries (Figure 2) [17]. The basic filtering of FGDE datasets was carried out by MS Excel and Orange software [23] to identify the optimized Kozak variants (Figure 1i). To visualize the distribution of Kozak variants and their enrichment coefficients, we used the scatter plot function of the Orange software package.

### 2.5. AAV Characterization

A quantitative assessment of AAV yield was carried out by qPCR using CMV-specific primers and an rAAV reference with a PCR titer pre-determined by Digital Droplet, as described earlier [15]. For stain-free PAAG gel, we loaded ~10^11^ viral particles and ran the gel for 2 h at 120 V. Protein bands were visualized after exposing the gel to UV_260_ for 2 min. To measure AAV infectivity, we transduced 50% confluent C12 cells with rAAV (MOI of 10^4^), followed by Ad5 infection (MOI of 5). At 48 h post-infection, cells were collected and analyzed by FACS. In the transduction verification experiment, we used original C12 cells without mCherry integration (Figure 1j).

## 3. Results

### 3.1. Cre Recombinase-Mediated DE Approach Workflow

A comprehensive study of the Kozak/AAV fitness landscape requires an assessment of two interrelated properties of AAV: structural fitness (assayed by a packaged virus yield) and infectivity, with both attributes dependent on the VP1 content within the capsid. Notably, the higher VP1 content renders AAV more infectious, but only up to a certain threshold, after which the packaging yield precipitously declines [15]. To account for both attributes’ relationship with both Kozak sequence libraries, (N)6ATGNN and (N)8ATGGC, we applied two orthogonal functional assays: packaging efficiency in Sf9 cells, and biological potency as assessed by infectivity in human C12 cells.

To identify Kozak variants with improved AAV transduction, we developed a Directed Evolution (DE) approach that relies on the Cre-mediated Lox inversion of the uncoated AAV DNA. This approach includes two main components: (1) AAV-Cap vector with Kozak sequence library (65,536 variants) linked in cis with Lox66/71 elements in a head-to-head orientation. In addition, the vector incorporated a GFP expression cassette for screening purposes. Both Kozak-Cap-Lox and GFP cassettes are placed between AAV ITRs for rescue/amplification, propagation, and packaging of the AAV in Sf9 cells; (2) A stable cell line based on C12 cells that expresses Cre-recombinase and provides Rep-mediated rescue, and Ad5-mediated ITR-cassette amplification after infection with rAAV (Figure 1, steps a, b). To incorporate Cre-recombinase, we used Tet-responsive vector carrying Cre and mCherry genes in cis-orientation.

Notably, during the construction of the pLox-Cap-KozakLib-EGFP vector, we found that the distance between Lox sites is critical for efficient Lox inversion; therefore, we introduced a spacer sequence to increase this by up to ~400 bp. While testing different conditions for infecting the HeLa C12Cre cell line with AAV2-Kozak library vectors, we identified the MOI of 10^2^ as the most reproducible (as compared to MOI of 10 or 10^3^). Using qPCR with primers to the original and inverted DNA sequences, we determined that applying the described protocol resulted in a 10^4^ enrichment of the inverted DNA sequences.

### 3.2. DE with Canonical (N)6ATGNN Library

Initially, we intended to identify Kozak variants with optimal transduction efficiencies that rely solely on standard multicycle-directed evolution-enhanced Cre-conversion. For this purpose, we used the Kozak library of (N)6ATGNN design and two cycles of DE in HeLa C12Cre cells. After selection, we sequenced 62 random separate clones and determined that two Kozak variants were detected in more than 75% of sequenced clones: Var#1 (TACTAT ATG CC, 50% of clones) and Var#2 (CGTTACATG_/_TG, with two consecutively deleted nucleotides after ATG start and ~26% of clones). The rest of the variants did not exceed 5% of the sequenced clones. We cloned Kozak Var#1 and Var#2 sequences into the pLox-Cap-KozakLib-EGFP vector and assembled AAV in Sf9 cells. To characterize the biological activity and VP–protein ratio of AAV-Var#1 and AAV-Var#2, we analyzed iodixanol-purified viral particles using stain-free PAAG. The analysis revealed that AAV-Var#1 had a VP1:VP2:VP3 capsid ratio comparable to AAV2 isolated from HEK293, with a somewhat higher VP1 content compared to the latter (Appendix A). The AAV-Var#2 capsid, on the other hand, consisted of VP2 and VP3 only, with no detectable VP1. The subsequent transduction assay confirmed the complete absence of infectivity for AAV-Var#2 compared to HEK293 AAV2 and AAV-Var#1. As each round of DE, by design, was dependent on the transduction of HeLa C12Cre cells, the enrichment of the non-infectious capsid variants such as AAV-Var#2 appears to be paradoxical. To explain this ambiguity, we supposed a “passenger/driver” relationship between different AAV variants, whereupon the less infectious but more represented variants (“passengers”) take a ride and escape from the endosome with the help of more infectious “drivers”. At the MOI of 10^2^ used at this step of the protocol, this free-ride phenomenon appears to preferentially favor the higher-yielding variants. Consistent with this notion, AAV-Var#2 was documented to be a highly robust capsid, yielding > 10^5^ capsids/cell. 

Since AAV-Var#1 had a favorable VP protein distribution, we expected that its biological activity would be similar to the HEK293-derived vector. Unfortunately, the leading variant (Var#1) demonstrated only 50% of the biological activity of the HEK293-derived vector. To account for this unexpected result, we assumed that the effect was caused by the alterations in the second amino acid in the VP1 protein of the selected Kozak motif vs. wild-type (Var#1 TACTAT-ATG CC vs. wild-type Kozak-ATG GC: Pro < Ala).

### 3.3. FGDE with (N)8ATGGC Library

In order to avoid the generation of AAV capsids with an altered second amino acid in the VP1 protein, we re-built the library, preserving the second codon wild-type sequence. Instead, we generated a Kozak library with the redundancy of all eight nucleotides upstream of the ATG start codon and GC as the second amino acid in the coding sequence: NNNNNNNN ATG GC, or (N)8ATGGC. To accelerate DE and to minimize the artifacts from a driver–passenger interaction, we introduced deep sequencing at the post-assembly stage, as well as a post-transduction libraries analysis. Furthermore, we added the FACS enrichment of transduced cells, only selecting for double-positive cell fractions: mCherry+/GFP+m where the former selects for Cre-positive cells and the latter identifies uncoated AAV DNA in the nuclei. The simultaneous usage of several powerful selection filters, such as Cre-mediated Lox inversion with the enrichment of transduced cell fraction and Rep rescue/amplification of uncoated DNA, all of these modifications significantly facilitated the identification of the optimal transducers after conducting only one cycle of AAV-Kozak DE. We dubbed this approach Fitness-Guided Directed Evolution (FGDE) because of its reliance on data filtering to extract pro-packaging motifs using both capsid assembly and transduction efficiencies (Figure 1).

### 3.4. Exploration of Packaging Landscape for (N)6ATGNN and (N)8ATGGC Kozak Libraries

The AAV production platform under investigation is based on insect Sf9 cells, and, as such, is not subject to the transcriptional control of the AAV genome in mammalian cells. The packaging efficiency in this system depends on the optimal ratios of the VP1:VP2:VP3 capsid and primarily relies on the transcription initiation rate from the start codon of VP1 [15]. To identify the optimal Kozak sequences directing the assembly of the most structurally fit capsids, we applied deep sequencing of the combinatorial Kozak libraries, both plasmid and assembled in viral particles. The comparative analysis of the variants’ distribution before and after assembly informs us of the effectiveness of a particular Kozak sequence as assessed by its fold change (dubbed packaging units).

To visualize the distribution of 65,536 variants and facilitate the extraction of the sequence-to-function relationship, we used a fractal-like DNA motif representation with a heat map (FDR-map) (Figure 2). A simple visual comparison of these maps for (N)6ATGNN-, and (N)8ATGGC libraries revealed a striking difference between the two, and the impact of the downstream Kozak sequence, affecting both the initiation rate and the composition of the amino acid following ATG. While (N)6ATGNN design incorporated a wide spectrum of sequences with a moderate packaging efficiency (~10 k variants), the (N)8ATGGC library generated a narrower packaging terrain with significantly higher peaks and more efficient packaging variants (pro-packagers, ~500 variants). It is documented that the A/T nucleotides downstream of the ATG start codon decrease the Kozak translation initiation efficiency which, in combination with the strong initiation motif upstream of ATG, generates a wider range of pro-packaging variants [17]. In a reciprocal manner, a strong initiation sequence downstream of ATG strengthens the relatively weak upstream Kozak combinations. In the case of (N)8ATGGC library design, to preserve a native amino acid sequence of VP1, we had to use strong GC sequences downstream the ATG start codon. This narrowed the set of potential good packagers upstream of the ATG to the Kozak variants with extremely low–moderate initiation efficiency.

Further, we extracted the most abundant packagers from the (N)8ATGGC library in silico, and identified pro-packaging motifs in Kozak sequences. The dominant motifs were as follows: NNNNATGN ATG GC, NNNNGTGN ATG GC, NNNNTTGN ATG GC, and NNNNTCGN ATG GC. All four motifs contained upstream, out-of-frame, alternative initiation codons, thus attenuating the strong downstream ATGGC content. Moreover, we established that pro-packagers within identified motifs displayed a higher packaging efficiency than the best packagers from the (N)6ATGNN library, and their out-of-frame preferences can be arranged in the following order: ATG > GTG > TTG ≥ CTG. These families can be described by a more general motif formula as (N)5TGN ATG GC. In addition, among pro-packager Kozak variants, we identified other variants of out-of-frame non-canonical codons upstream of the ATG start codon, such as (N)4ACGNATGGC.

### 3.5. Exploration of Transduction Landscape for (N)8ATGGC Kozak Library

Considering that modifying the second amino acid sequence of VP1 might affect AAV transduction even in case of the optimal VP1:VP2:VP3 protein distribution (1:1:10), we decided to further explore only the N(8)ATG GC Kozak library using the FGDE approach. To assess the transduction efficiencies, we infected HeLa CreC12 cells with the packaged library at the MOI of 10^2^. Infected cells were enriched by FACS sorting, and Cre-Lox-inverted genomes were amplified by PCR with the respectively designed primers using the total genomic DNA isolated from the enriched cell population. The amplified inverted DNA was then subjected to the NGS analysis. Notably, we performed FGDE screening in two independent biological replicates at each stage of the experiment, starting from separate cell lines batches. Because Kozak motif’s composition significantly affects both AAV packaging ability and the variants’ representation in the assembled AAV library, we introduced a normalization factor for the assessment of transduction enrichment: the ratio of a given Kozak variant NGS reads in the inverted DNA (which is a proxy for the transducing variants), and the same Kozak reads in the packaged AAV library (dubbed infectivity units). After normalization, we ascertained that the vast majority of super producers have relatively low transduction efficiencies. That is not surprising as VP1 carries phospholipase A2, facilitating AAV egress from the late endosomes, and that VP1 content is inversely correlated with the efficiency of the capsid assembly. Therefore, we did not document clear pro-transduction peaks in the transduction scatter plot in the area that correspond to TGNATGGC families (Appendix A). On the contrary, we detected the appearance of super-transducers with significant enrichment of inverted nuclear AAV DNA and a low abundance in the original assembled AAV Kozak library. Unfortunately, these super-transducers were characterized by an extremely low yield and their further study was not practically possible. The presence of the variants demonstrating extremely high infectivity enrichment and a pronounced low yield (and thus low industrial value) can be explained by biological noise due to the high-throughput nature of the screening.

### 3.6. Identifying Optimal Kozak Variants (Packagers/Transducers) Based on Multistep Filtering Workflow

In order to establish a pipeline for an analysis of Kozak distributions into packaging-infectivity dimension, we used a logarithmic representation of the data. We conducted an unsupervised clustering analysis to identify separate clusters based on a k-means approach with the highest Silhouette coefficient (Figure 3). For subsequent analysis, we selected cluster C1, which represents samples with a balanced distribution of packaging–infectivity values. The omission of C2 and C3 was caused by the potential nature of Kozak from these clusters: C2 represents hypothetical super-transducers with neglectable packaging, and C3 represents super-producers with little transduction activity. Further, we applied the filtering criteria to narrow down the set of Kozaks to the most potentially effective, with balanced packaging–infectivity properties. We cut off samples with an excessive coefficient of variation (CV) between two biological replicates in transduction assays (infectivity units, 1/CV < 3), and samples with too little packaging (<5 packaging units) and too low transduction activity (<1 infectivity unit). Overall, we identified 109 potential Kozak variants for subsequent routine analysis (Appendix A).

### 3.7. Characterization of the Selected Kozak Variant by VP Composition, Transduction Efficiencies, and Yield

The FGDE approach provides us with only packaging and transduction (infectivity) enrichments, but not with actual yields per cell or transduction efficiences at a certain MOI. This approach provides us with leads rather than with precise information regarding the most optimal variants, which should be viewed as a wide, continuous range of changing packaging/transduction relationships. Therefore, we randomly selected several variants with a higher yield and moderate transduction-enrichment coefficients to verify their absolute packaging/infectivity characteristics and to create reference points for the subsequent classification of the most promising variants. All four selected Kozak variants belonged to the ATGNATG subfamily and demonstrated yield in the range from ~25,000 to ~52,000 encapsidated vector genomes per Sf9 cell, measured by qPCR. Stain-free PAAG assay showed an abnormally elevated level of VP1 relative to VP2 and VP3 proteins in packaged, iodixanol-purified viral particles: VP2/VP1 in the range 0.3–0.6 and VP3/VP1 in the range 9–14 (Table 1A, Appendix A). The subsequent transduction assay with iodixanol-purified AAV-Kozak revealed that one out of four variants (Var#3) had an infectivity equal to HEK293-derived AAV2. The rest of the variants (3 out 4) demonstrated ~70% of the activity of HEK293 AAV2 (Table 1B, Appendix A).

Using data from the routine characterization of the selected variants and FGDE-enrichment coefficients, we investigated the correlation of standard vs. high-throughput analyses. Both plots, qPCR yield vs. packaging enrichment and FACS infection assays vs. FGDE transduction enrichments, demonstrated a high level of correlation (~0.89) between standard and high-throughput data (Appendix A). Moreover, the Kozak variant with the highest transduction efficiency among the four pre-selected variants according to qNGS data demonstrated the same activity as HEK293 AAV2. These results emphasize the high predictive power of the FGDE approach for the accelerated one-cycle selection of the most optimal AAV packager/transducer for the Sf9 production system.

## 4. Discussion

Recently, we described a OneBac3.0 Sf9 system with empirically identified, attenuated, serotype-specific Kozak sequences that produced rAAV5 and rAAV9 with increased VP1/VP2 content, exceptional DNA purity, and improved transduction efficiency [15]. Such empirical screening within a set of 65,536 potential sequences using the interdependent attributes of structural fitness and biological potency is an extremely laborious proposition. Moreover, despite the seemingly optimal VP composition (1:1:10), as assessed by protein gel analysis in the cell crude lysate, some iodixanol-purified viruses with empirically modified Kozak sequences demonstrated sub-optimal transduction efficiencies [15]. The current strategy, therefore, is an attempt to develop a serotype-independent high-throughput approach to screening 65,536 Kozak sequences for the vectors manufactured in the Sf9 system.

At the beginning, we applied the standard DE of combinatorial Kozak rAAV libraries using multicycle enrichment, expecting to select the variants optimized for both transduction and packaging. To increase selection pressure, we included Cre-Lox inversion as a marker of successful rAAV transduction since it required viral particle uncoating and dsDNA synthesis. For this purpose, we combined cis-orientation (N)6ATGNN motifs with Lox-elements and performed two rounds of selection in the Cre-expressing cell line. We also hypothesized that mutating the second amino acid in the VP1 in the (N6)ATGNN might be associated with alterations in VP1 function. Indeed, a seemingly ideal 1:1:10 VP composition for the selected Var#1 capsid manifested in only 50% of the WT infectivity rate. Notably, we observed a similar effect for AAV5 and AAV9 Kozak adjustments when the selected variants with close to 1:1:10 VP protein distribution demonstrated reduced infectivity as compared to their WT serotypes [15]. Another complication for the original DE protocol was the effect of the driver—passenger interaction when selecting for the VP1-less non-infectious virus-like particles. To overcome the apparent setback, we introduced a high-throughput approach based on NGS and bioinformatic analysis, a Fitness-Guided Directed Evolution. Instead of a straightforward physical selection in Sf9 and HeLa C12 cells, the FGDE utilizes the NGS dataset for the original plasmid and viral libraries. Comparing both, we derived an enrichment coefficient and identified Kozak sequences affecting the structural fitness, both at the level of VP1:VP2:VP3 capsid ratios and also accounting for a possible VP1 second amino acid substitution effect on packaging. By the same token, using Kozak sequence distributions in the assembled viral library vs Lox-flipped dsDNA from positively transduced HeLa cells, the enrichment coefficient was also calculated for the transduction efficiency attribute.

First, we compared the packaging landscapes of (N)6ATGNN and (N)8ATGGC Kozak libraries under identical conditions in Sf9 cells. The respective heat maps of these two libraries showed a dramatic difference between the two libraries, with a broad range of Kozak sequences for the former and the narrow packaging “ridges” for the latter, which were generally associated with the common motif of (N)4TGNATGGC. At first glance, the wasteful upstream initiation should deplete the total ribosome pool and decrease the final capsid production, including the major in-frame protein, VP3. However, a closer analysis reveals that AAV2 and the vast majority of other AAV serotypes incorporate out-of-frame stop codons very close to the alternative out-of-frame initiation codons in their VP1 upstream sequence. This facilitates greater rates of premature ribosome release from the non-target short mRNA transcript, preserving ribosomal potential to re-initiate translation from downstream VP2 and VP3 start codons (Appendix A) [24].

Assessing the transduction landscape is even more challenging because the efficient tracing of transducer clones relies on two parameters: a quantifiable amount in the assembled library, and an efficient transduction. More often than not, the most efficient transducers, which, according to qNGS, had the highest enrichment factor, were not reproducible in biological replicates. Moreover, such variants demonstrated barely traceable coverage in the assembled rAAV-Kozak library. Several attempts to accumulate such Kozak variants in amounts sufficient for validation were unsuccessful. Therefore, we had to modify the filtering protocol, significantly improving a selection of transducers with acceptable capsid yield.

Previously, we observed a direct correlation between VP1 content and the virus’ infectivity, but an inverse correlation with its structural fitness manifesting in capsid productivity [15]. Thus, upon DE selection, higher VP1 (infectivity) is accompanied by lower yield, and higher capsid yield by lower infectivity. This results in the selection of multiple “good” capsid variants with low-to-moderate transduction and above-average yield. In other words, we cannot technically identify the most efficient transducer by straightforward selection due to the need to rely on two orthogonal attributes: yield and biological potency. In the FGDE approach, we are able to read both parameters separately (yield/infectivity) and pick the variant with suitable features (Figure 4). The large-scale experiment revealed a transduction saturation pattern with an insignificant inverse correlation between packaging and infectivity (R = −0.2) (Appendix A).

The FGDE, as described, is a universal approach that is suitable for the adjustment of the packaging/transduction properties of rAAV capsids for multiple serotypes in the Sf9 system. The transferability of the approach to other rAAV serotypes can be demonstrated by following the described protocol. The FGDE approach provides the set of enrichments for a wide range of variants, but not direct yield/infectivity values. If the optimized Kozak sequence for the alternative rAAV serotype is placed within the coordinates of the packaging/infectivity of FGDE, the approach should enable to narrow down a range of necessary enrichments and identify variants with the same optimized infectivity. In FGDE coordinates, the optimized AAV9 Kozak variant–CTTAGTGTATGGC [15]—has an infectivity enrichment of about 1 and packaging of ~10 coefficients. Therefore, we picked four Kozak variants with close to CTTAGTGTATGGC enrichment coefficients, but completely different structures and better yields: AACGATGCATGGC, CAACATGAATGGC, GATCATGGATGGC, CGCGATGGATGGC. One of them, Var#3, is reminiscent of the infectivity of the optimized AAV9 insect Kozak vs. AAV9 HEK293, despite significant structural discrepancies and improved yield [15]. Among the rest of the 109 pre-selected by FGDE variants, we detected even better transducers with moderate yield and stable reproduction of transduction assays according to the FGDE approach (Appendix A).

We would like to emphasize the high correlation between FGDE enrichments and the biological properties of the capsids that we identified on pre-selected Kozak variants (Appendix A). These properties of the FGDE approach can be utilized for the specific industrial adjustment of capsid features when someone requires more yield from an economical standpoint or high infectivity to reduce the dose per kg upon gene therapy.

## 5. Conclusions

The FGDE approach identified a set of potential packagers/transducers for AAV2 production in Sf9 cells using the principle of leaky ribosomal scanning implemented in the OneBac3.0 platform. The same approach could be potentially utilized serotype-independently for the identification of optimized producers of alternative AAV serotypes in Sf9 cells.

## 6. Patents

This work resulted in an international patent application: PCT/US2021/029749, WO/2021/222472.

## Figures and Tables

**Figure 1 viruses-15-01983-f001:**
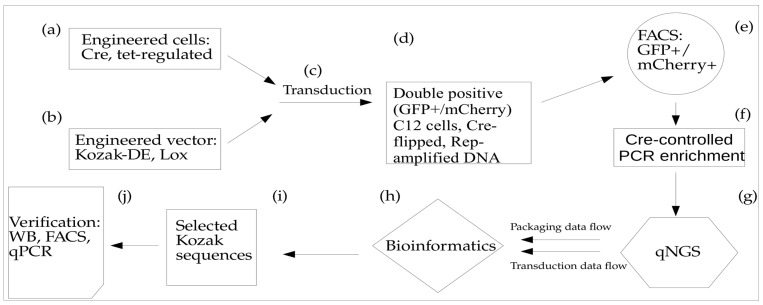
Schematic workflow of FGDE for (N)8ATGGC AAV library in Sf9 insect system. Stages of FGDE approach: (**a**)—cell line construction, (**b**)—vector engineering, (**c**)—transduction of Cre-expressing cells by AAV library, (**d**)—Cre-inversion inside of transduced cells, (**e**)—FACS sorting, (**f**)—PCR amplification stage, (**g**)—measuring of Kozak distribution; (**h**)—Kozak characterization and selection, (**i**)—data filtering, (**j**)—routine characterization of the selected Kozak variants.

**Figure 2 viruses-15-01983-f002:**
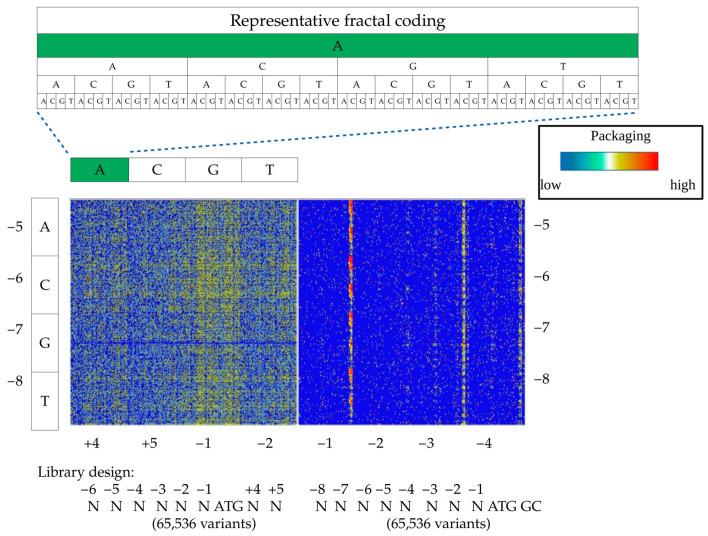
Fractal representation of packaging of two Kozak AAV2 libraries according to qNGS data.

**Figure 3 viruses-15-01983-f003:**
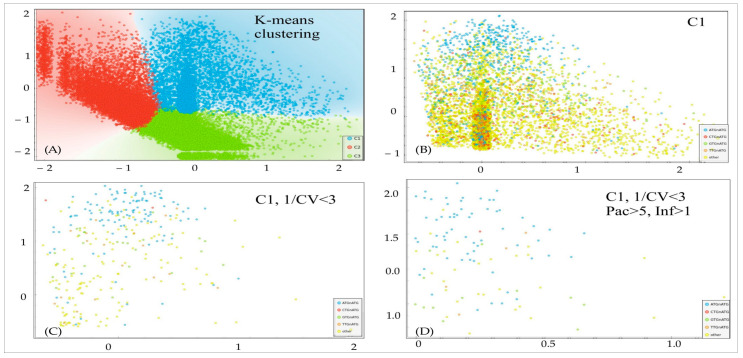
Selection of Kozak variants with optimal packaging-infectivity rates based on FGDE data. Axis X—Log10 (average transduction). Axis Y—Log10 (packaging). (**A**) C1—blue cluster, C2—red cluster and C3—green cluster. (**B**) Motifs inside the C1 cluster: blue—NNATGNATGGC, red—NNCTGNATGGC, green—NNGTGNATGGC, orange—NNTTGNATGGC and yellow—other. (**C**) Variants with highly reproducible infectivity in two replicates. (**D**) Variants with highly reproducible infectivity and substantial infectivity/packaging efficiencies.

**Figure 4 viruses-15-01983-f004:**
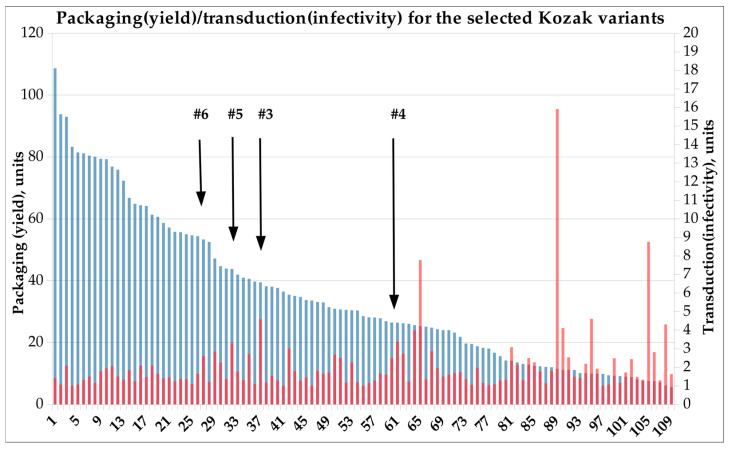
Double Y axes graph of packaging and infectivity for pre-selected Kozak variants. Based on FGDE packaging and infectivity enrichments. Blue bars—packaging units, red—transduction units.

**Table 1 viruses-15-01983-t001:** Characterization of the selected Kozak variant. (**A**) density analysis of VP protein distribution in stain-free gel; (**B**) FACS-sorting infectivity, MOI 10,000.

**A**					
**Kozak**	**VP Ratio Relative to VP1**			**Kozak Sequence**	
	**VP1**	**VP2**	**VP3**		
AAV-Var#3	1.0	0.3	9.0	AACGATGC ATG GC	
AAV-Var#4	1.0	0.3	8.0	CAACATGA ATG GC	
AAV-Var#5	1.0	0.6	9.0	GATCATGG ATG GC	
AAV-Var#6	1.0	0.4	14.0	CGCGATGG ATG GC	
**B**					
**rAAV2**	**Repl.#1, %**	**Repl.#2, %**	**Repl.#3, %**	**Aver., %**	**St.Dev** **.**
AAV-Var#3	25.1	26.7	25.3	25.7	0.9
AAV-Var#4	20.0	18.2	16.3	18.2	1.9
AAV-Var#5	20.0	17.0	19.8	18.9	1.7
AAV-Var#6	15.4	16.9	19.5	17.3	2.1
HEK293	28.6	25.3	22.4	25.4	3.1

## Data Availability

The data presented in this study are available on request from the corresponding author. The data are not publicly available due to commercial interest.

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
