# Peer review of "Exploring the Comprehensive Kozak Sequence Landscape for AAV Production in Sf9 System"

_viruses, 2023, doi:10.3390/v15101983_

Round 1

Reviewer 1 Report

Major:

The authors demonstrated the design and application of an NGS-driven selection system for a Cap2 Kozak sequence intended to improve capsid protein ratios in the OneBac AAV production system. Authors successfully cross-validated lead candidates in the same system. However, the Cap gene is amplified and packaged in the presented selection system, whereas it is stably integrated into the Sf9 genome in the OneBac system, potentially leading to different expression patterns of VP proteins between the two systems. Therefore, I can not recommend publication of this article until the authors have demonstrated the transferrability of the results to the OneBac platform.

Minor:

The first paragraph of the introduction seems outdated. We now have seven approved AAV-based therapeutics so AAV-based therapies clearly have moved beyond "demonstrat[ing] promising potential".

It is not clear how the libraries were diversified. I assume PCR with degenerate primers?

The authors are kindly asked to include the results described in lines 170 - 176. At the least should they annotate the primer locations in Figure S1.

qNGS is sometimes used instead of NGS but for the sake of clarity, I find NGS to be sufficient and certainly more common.

The authors have chosen to co-infect C12 cells with Ad5 for their AAV infectivity assay. Co-infection should not be necessary to investigate transduction efficiency and the authors are kindly asked to provide an explanation.

Authors seem to have chosen an arbitrary fluorescence threshold in their infectivity assay (Figure S5), leading to a relatively low reported transduction efficiency. Typically, the threshold is set to 1% transduction in the untransduced control cells (which would work in the authors favour).

Figure 2 should include the nucleotide positions marked on the axis (going down the fractal), otherwise the figure is ambiguous. Can the "packaging" color scale be referenced to something, e.g., packaging from an unmodified Kozak progenitor construct?

The labels in Figure 3 are too small.

The authors' claim of packaging efficiency depending on the capsid protein ratios is counterintuitive and should be explained in more detail (lines 226ff, also in the abstract). The immediate results presented show a negative correlation between packaging and transduction which might be more easily explained by an overall higher capsid yield if mainly VP3 is expressed.

The title including "machine guided" implies involvement of some kind of "decision-making machine" or AI in the study, however all decision making is done by the human investigator. Maybe "data driven" would be the more appropriate term.

Square brackets and parentheses are used interchangibly in references to figures and tables, please correct (e.g. line 403).

Please check grammar throughout the manuscript, especially usage of indefinite articles.

The author contributions still contain remainders of the word template.

Author Response

Dear Reviewer,

Thank you very much for taking the time to review this manuscript. Please find the detailed responses below. The corresponding revisions/corrections are highlighted in the manuscript.

Major:
The authors demonstrated the design and application of an NGS-driven selection system for a Cap2 Kozak sequence intended to improve capsid protein ratios in the OneBac AAV production system. Authors successfully cross-validated lead candidates in the same system. However, the Cap gene is amplified and packaged in the presented selection system, whereas it is stably integrated into the Sf9 genome in the OneBac system, potentially leading to different expression patterns of VP proteins between the two systems. Therefore, I can not recommend publication of this article until the authors have demonstrated the transferrability of the results to the OneBac platform.

Author Response: We would like to express our gratitude for highlighting this aspect of our work. Hereby, we indeed used transient expression approach (pLox-Cap-KozakLib-EGFP vector)  for accelerated screening of 65,536 Kozak variants in Sf9 system. In this respect, it would be difficult to imagine screening so many variants in the stable integration mode. Although our current experimental system is based on transient transfection, and OneBac is based on stable cell lines (containing Rep and Kozak-Cap cassettes) thus potentially affecting total amount of Cap mRNA in the cell, both systems have identical promoter, coding sequence and downstream translation organization. In both systems, VP1, VP2 and VP3 proteins translations are carried out from a single polycistronic mRNA (Urabe et al., 2002, Aslanidi et al., 2009) with identical 5’-UTR organization and utilizing the same identical ATG-skipping mechanism translation of Cap mRNA. Therefore, integration of  pLox-Cap-KozakLib-EGFP into genome would not affect stoichiometry of VP1, VP2, and VP3 protein distribution that defines packaging and infectivity. Moreover, we would even argue that because of the stochasticity of the integration process in OneBac system, the random nature of the integration “landing pad”, and the different number of the integrated copies, studying the transient expression of the plasmid in Sf9 is more salient when comparing it to the transient expression from the plasmid in HEK 293 system. 

Minor:
The first paragraph of the introduction seems outdated. We now have seven approved AAV-based therapeutics so AAV-based therapies clearly have moved beyond "demonstrat[ing] promising potential".

Author Response: We updated reference on the most recent: “There are multiple ongoing pre-clinical/clinical trials of rAAV gene therapy and seven of them are approved by FDA/EMA: Glybera, Luxturna, Zolgenesma, Hemgenix, Roctavian, Upstaza and Elevidys  [2]”.

It is not clear how the libraries were diversified. I assume PCR with degenerate primers?

Author Response: To clarify details of experiments with library diversification, we added the following text:
“In order to synthesize 65,536 Kozak variants, we used degenerate oligos with theoretically equal distribution of four possible nucleotides among 8 positions: NNNNNNNN – 48 = 65,536 combinations”.

The authors are kindly asked to include the results described in lines 170 - 176. At the least should they annotate the primer locations in Figure S1.

Author Response: To clarify details of experiments in lines 170-176, we added the following text and annotated the primer location in Figure S1: 
“To carry out Cre-inverted PCR, we used invFor/wtRev - Polh-Rev primers which had Rev-Rev orientation before Cre-mediated inversion and For-Rev orientation in Cre-inverted DNA (Fig. S1 (pink).  Subsequent PCR barcoding was performed by 8 cycles amplification of gel-purified Cre-inverted PCR product with internal NGS-For/Rev (Fig. 1S (grey)) primers”. 

qNGS is sometimes used instead of NGS but for the sake of clarity, I find NGS to be sufficient and certainly more common.

Author Response: Here, we used ultra-deep coverage that exceed 300x for measuring of Kozak variants distribution in AAV library and transduced cells. The term quantitative next generation sequencing (qNGS) was used by several authors to emphasize NGS with >50x coverage (PMC5042788 or PMC9289621). We added explanation in the text:
“In order to comprehensively characterize Kozak variant distributions, we used ultra-deep coverage NGS (>300x) at the most diverse stage - plasmid library. Therefore, we dubbed it quantitative NGS (qNGS)”.

The authors have chosen to co-infect C12 cells with Ad5 for their AAV infectivity assay. Co-infection should not be necessary to investigate transduction efficiency and the authors are kindly asked to provide an explanation.

Author Response: Although the reviewer is absolutely correct regarding AAV potential to infect cells alone without Ad5, we will explain critical necessity of Ad5 co-infection usage in the current project. We used Ad5 to stimulate double-stranded DNA synthesis upon delivery of single-stranded ITR-Lox-Cap-KozLib-EGFP cassette. Upon co-infection, Ad5 provides helper functions for AAV to more rapidly convert ssDNA AAV into dsDNA [PMC7354565]. Otherwise, the process of dsDNA synthesis and Cre-mediated inversion would take several weeks post-infection. To increase the stringency of the selection, it was necessary to use low MOI resulting in low numbers of positive cells that did not exceed 10% (before FACS enrichment), therefore the long incubation of cells incubation after infection with ssAAV would lead to a significant dilution of positive cells and abrogation of the sensitivity. In case of transduction comparison for selected Kozak (4 variants), we used Ad5 co-infection because we also packaged ssDNA cassette ITR-Lox-Cap-Koz-EGFP. Such approach is standard in case of testing ssAAV infectivity and was used by many authors [PMC2687429, PMC3955967].

Authors seem to have chosen an arbitrary fluorescence threshold in their infectivity assay (Figure S5), leading to a relatively low reported transduction efficiency. Typically, the threshold is set to 1% transduction in the untransduced control cells (which would work in the authors favour).

Author Response: Although using 1% transduction threshold would work in our favor, the decision of a threshold value was dictated by the concern of a collecting false-positive auto-fluorescent cells in the transduced cells pool thus reducing the stringency of the selection.  

Figure 2 should include the nucleotide positions marked on the axis (going down the fractal), otherwise the figure is ambiguous. Can the "packaging" color scale be referenced to something, e.g., packaging from an unmodified Kozak progenitor construct?

Author Response: We added axis to specify nucleotide position. To our knowledge, such unmodified Kozak progenitor for insect cells has not been described. The standard Kozak consensus sequences are usually represented by degenerate sequences: for vertebrates – gccRccATGG, and for a D. melanogaster – atMAAMATGamc. In our opinion, therefore, presentation of the data in question in correlation to the unrelated system is not informative.

The labels in Figure 3 are too small.
Author Response: Corrected

The authors' claim of packaging efficiency depending on the capsid protein ratios is counterintuitive and should be explained in more detail (lines 226ff, also in the abstract). The immediate results presented show a negative correlation between packaging and transduction which might be more easily explained by an overall higher capsid yield if mainly VP3 is expressed.

Author Response: We have recently explored the relationship between the VP1/2/3 stoichiometry, virus packaging potential, and its biological potency [PMC5768557]. Moreover, other groups investigated dependency of AAV fitness and VP1 modifications revealing that exclusion of VP1 PLA domain from internal localization increases AAV packaging. In other words, compaction of VP1 to the size VP2/VP3 leads to better packaging fitness [PMID: 37131661]. On the other hand, the critical role of VP1 (specifically, its VP1up PLA2 domain) in AAV infection is a well described phenomenon, and the elimination/depletion of VP1 leads to abrogated AAV infectivity [PMC3416132, PMC3955967]. Summarizing, high VP1 content favors higher infectivity but complicates assembly, while low VP1 content/no VP1 – favors higher capsid yield and low/absent AAV infectivity. 

Comments on the Quality of English Language

The title including "machine guided" implies involvement of some kind of "decision-making machine" or AI in the study, however all decision making is done by the human investigator. Maybe "data driven" would be the more appropriate term.

Author Response: We modified the title to avoid directly implying ML involvement.

Square brackets and parentheses are used interchangibly in references to figures and tables, please correct (e.g. line 403).
Author Response: Corrected.

Please check grammar throughout the manuscript, especially usage of indefinite articles.
Author Response: Done

The author contributions still contain remainders of the word template.
Author Response: Corrected.

Sincerely,

Oleksandr

Reviewer 2 Report

Kondratov and Zolotukhin.  Machine-guided directed evolution of rAAV combinatorial libraries in Sf9 insect cells.

The research article entitled, “Machine-guided directed evolution of rAAV combinatorial libraries in Sf9 insect cells” describes a library screening approach that identified an optimized Kozak sequence to explicitly express the VP1 gene of AAV serotype 2 (AAV2) in Sf9 insect cell lines within the OneBac3.0 expression system.  This study is important for the gene therapy field, specifically for AAV vector production, since improper ratios of expression between the VP1, VP2, and VP3 genes in Sf9 producer cells often lead to production issues and vector potency problems in tissues/cells of the patient.  The authors initially used a Kozak library designed as (N)6ATGNN to screen for optimal sequence variants.  This library was first tested in HeLA C12 production cells, resulting in two predominant sequences after two rounds of selection (i.e., directed evolution, DE).  One variant showed a complete loss of transduction.   The authors surmised that the loss of transduction was due to a conversion of the second amino acid from alanine to proline.  Therefore, a second library was synthesized (N)8ATGGC and screened along with the (N)6ATGNN library in Sf9 producer cell lines in one round.  Since the second library was guided by information gleaned from the initial screen, this method was coined, “machine-guided directed evolution (MGDE)”.  The top variants identified by the MGDE method revealed that ATGNATG Kozak motifs were the most effective for AAV2 production in Sf9 cells.  Four variants were highlighted in the study and tested for their capacity to package vector and transduce cells in culture.  One Kozak variant (Koz/Var #3) led to vectors that transduced equally as well as HEK293-packaged vectors.

Overall, this reviewer anticipates that this report will be of interest to the AAV/gene therapy research community.  The authors are experts in this field.  The work and findings also have commercial value, as indicated by the author’s international patent application disclosure on this technology.  However, the manuscript is marked by multiple reporting issues that require addressing.  There are some grammar issues, inconsistencies in the annotation of candidate variants, and a lack of description in some areas that keep this study from publication.  The paper requires a round of polishing before recommendation for acceptance.  In addition, this reviewer feels that the authors have taken liberties with the term “machine-guided” and “directed evolution”.  These key points are described in detail as bullet points below.  In summary, this reviewer recommends acceptance of the manuscript to Viruses following revisions.

Major points:

·       In this reviewer’s opinion, the author’s uses of the terms “directed evolution” and “machine-guided” are not correct.  According to the methods, only two rounds of selection were performed using the (N)6ATGNN library in the C12 cells.  After the first round of selection, the library was PCR-amplified and re-cloned back into plasmids for the second round of screening.  In between these two rounds, there is no indication of mutagenesis (based on the methods), which is a hallmark of library diversification and a key step in the directed evolution process.  Also, the decision to move from the (N)6ATGNN to the (N)8ATGGC seems to be based on the sequencing of 62 random clones that yielded two predominant clones, one of which failed to transduce cells as a result of an alanine-to-proline mutation in the second amino acid of VP1.  It is not clear to the reviewer in what way computational guidance was used.

Furthermore, the MGDE approach was done using only one round of selection; therefore, it does not fit the definition of directed evolution.  Thus, the first screen performed by the authors (section 3.2) would be best described as an “iterative selection” of optimized Kozak motifs; whereas the second screen (section 3.4) would be best described as a “fitness-guided single-round selection” of optimized Kozak motifs.

·       In lines 181 to 184, the authors introduce Var#1 and Var#2.  The reviewer feels that it is important to disclose what these candidate sequences are.  Are these the same top candidates observed in the (N)8ATGGC library?  Also, there is inconsistency with the naming.  In the manuscript text and tables, these are primarily referred to as Var #1-6, but in the figures, they are referred to as Koz #1-6.  It is recommended to keep the same nomenclature throughout the manuscript.

·       There is no mention in the main text of the data presented in Table 1C.  Additionally, it is not clear what “Inf.Enr.”, “Pack.Enr.” mean, and what the units are for these values.

·       In this reviewer’s opinion, Figure S8 is both very interesting and important.  It is recommended to elevate this data to a main figure.  It is also recommended to not stack the transduction values on top of the packaging values, as this masks the values from variant #81 onward.

Minor points:

·       The manuscript requires proof reading to correct grammar errors.  The following are a few examples:

o   Line 74: Please revise to: “Using this approach allowed us to narrow down the most optimized…”

o   Line 99: The motif only shows five Ns (NNNNNATGNN), not six.

o   Lines 109-110: Please revise to: “In this way, Kozak sequences of successfully transduced variants were PCR-amplified and re-cloned into the…”

o   Lines 179-180: Please revise to: “Initially, we intended to identify Kozak variants with optimal transduction efficiencies that rely solely on standard multicycle-directed evolution enhanced Cre-conversion”

o   Line 249: Please revise to: “…we had to use strong GC sequences…

o   Lines 253-261: The authors describe 4 motifs: NNNNATGNATGGC, NNNNGTGNATGGC, NNNNTTGNATGGC, and NNNNTCGNATGGC.  However, the following text in line 261 refers to ATG>GTC>TTG³CTG.  GTC and CTG do not match those described previously.  Please resolve or clarify.

o   Line 452: Please revise to: “This work has resulted in an international patent application: PCT/US2021/029749…”

o   Line 476: Please revise to: “We thank the UF UCBR NGS facility…”

Author Response

Dear Reviewer,

Thank you very much for taking the time to review this manuscript. Please find the detailed responses below. The corresponding revisions/corrections are highlighted in the manuscript.

Open Review
(x) I would not like to sign my review report 
( ) I would like to sign my review report 
Quality of English Language
( ) I am not qualified to assess the quality of English in this paper 
( ) English very difficult to understand/incomprehensible 
( ) Extensive editing of English language required 
(x) Moderate editing of English language required 
( ) Minor editing of English language required 
( ) English language fine. No issues detected 

                    Yes     Can be improved     Must be improved     Not applicable
Does the introduction provide sufficient 
background and include all relevant references?
                    (x)         ( )             ( )             ( )
Are all the cited references relevant 
to the research?
                    (x)         ( )             ( )             ( )
Is the research design appropriate?
                    ( )         (x)             ( )             ( )
Are the methods adequately described?
                    ( )         (x)             ( )             ( )
Are the results clearly presented?
                    ( )         (x)             ( )             ( )
Are the conclusions supported by the 
results?
                    (x)         ( )             ( )             ( )
Comments and Suggestions for Authors
Kondratov and Zolotukhin.  Machine-guided directed evolution of rAAV combinatorial libraries in Sf9 insect cells.

The research article entitled, “Machine-guided directed evolution of rAAV combinatorial libraries in Sf9 insect cells” describes a library screening approach that identified an optimized Kozak sequence to explicitly express the VP1 gene of AAV serotype 2 (AAV2) in Sf9 insect cell lines within the OneBac3.0 expression system.  This study is important for the gene therapy field, specifically for AAV vector production, since improper ratios of expression between the VP1, VP2, and VP3 genes in Sf9 producer cells often lead to production issues and vector potency problems in tissues/cells of the patient.  The authors initially used a Kozak library designed as (N)6ATGNN to screen for optimal sequence variants.  This library was first tested in HeLA C12 production cells, resulting in two predominant sequences after two rounds of selection (i.e., directed evolution, DE).  One variant showed a complete loss of transduction.   The authors surmised that the loss of transduction was due to a conversion of the second amino acid from alanine to proline.  Therefore, a second library was synthesized (N)8ATGGC and screened along with the (N)6ATGNN library in Sf9 producer cell lines in one round.  Since the second library was guided by information gleaned from the initial screen, this method was coined, “machine-guided directed evolution (MGDE)”.  The top variants identified by the MGDE method revealed that ATGNATG Kozak motifs were the most effective for AAV2 production in Sf9 cells.  Four variants were highlighted in the study and tested for their capacity to package vector and transduce cells in culture.  One Kozak variant (Koz/Var #3) led to vectors that transduced equally as well as HEK293-packaged vectors.
Overall, this reviewer anticipates that this report will be of interest to the AAV/gene therapy research community.  The authors are experts in this field.  The work and findings also have commercial value, as indicated by the author’s international patent application disclosure on this technology.  However, the manuscript is marked by multiple reporting issues that require addressing.  There are some grammar issues, inconsistencies in the annotation of candidate variants, and a lack of description in some areas that keep this study from publication.  The paper requires a round of polishing before recommendation for acceptance.  In addition, this reviewer feels that the authors have taken liberties with the term “machine-guided” and “directed evolution”.  These key points are described in detail as bullet points below.  In summary, this reviewer recommends acceptance of the manuscript to Viruses following revisions.

Major points:
·       In this reviewer’s opinion, the author’s uses of the terms “directed evolution” and “machine-guided” are not correct.  According to the methods, only two rounds of selection were performed using the (N)6ATGNN library in the C12 cells.  After the first round of selection, the library was PCR-amplified and re-cloned back into plasmids for the second round of screening.  In between these two rounds, there is no indication of mutagenesis (based on the methods), which is a hallmark of library diversification and a key step in the directed evolution process.  
Although Directed Evolution (DE) approach has multiple technical flavors and variations, including in-between rounds of mutagenesis/shuffling/etc (re-diversification), AAV community often uses the broad term DE as applied to the capsid selection in iterative evolution conducted in many different ways [Xiao Xiao et al., 2021 (PMC7773954); R Jude Samulski et al., 2009 (PMC2801879), Sabeti et al., 2021 (PMID: 34506722), Minmin Luo et al., 2022 (PMID: 35879607), Marsic et al., 2014 (PMID: 25048217). True, the current project doesn’t modify capsid per se, just their stoichiometry. Here, we’re using the term DE to avoid unnecessary introduction of the novel terminology regarding such a familiar for AAV community process, and therefore, we would like to retain DE term for (N)6ATGNN library selection even though the described protocol is somewhat different, with all due respect to reviewer’s notion. On the other hand, we agree with the reviewer on the point of “machine-guided” (or rather the absence of) and propose the alternative title.
Also, the decision to move from the (N)6ATGNN to the (N)8ATGGC seems to be based on the sequencing of 62 random clones that yielded two predominant clones, one of which failed to transduce cells as a result of an alanine-to-proline mutation in the second amino acid of VP1.  It is not clear to the reviewer in what way computational guidance was used.
Computational guidance was used to analyze NGS datasets for the packaging fitness libraries, both (N)6ATGNN and (N)8ATG, as well for the transduction efficiency in (N)8ATG library. Sixty-two Sanger sequencing of the (N)6ATGNN library directed us to modify the Kozak’s sequence from (N)6ATGNN to the (N)8ATGGC.

Furthermore, the MGDE approach was done using only one round of selection; therefore, it does not fit the definition of directed evolution.  Thus, the first screen performed by the authors (section 3.2) would be best described as an “iterative selection” of optimized Kozak motifs; whereas the second screen (section 3.4) would be best described as a “fitness-guided single-round selection” of optimized Kozak motifs.
We agree with the reviewer that MGDE approach doesn’t exactly fit the definition of machine guidance and directed evolution. Therefore, we would like to modify the term “Machine-Guided” for Fitness-Guided Directed Evolution (FGDE).

In lines 181 to 184, the authors introduce Var#1 and Var#2.  The reviewer feels that it is important to disclose what these candidate sequences are.  Are these the same top candidates observed in the (N)8ATGGC library?  

We added  Var#1 and Var#2 sequences: TACTATATGCC and  CGTTACATG_/_TG (two  deletions that abrogated VP1 translation), respectively. As we mentioned Var#1 has  alanine-to-proline mutation, therefore, to avoid VP1 protein mutation, we “froze” wild type sequence in (N)8-ATG-GC design (N8-Met-Ala, GCT – is Ala codon) vs (N)6ATGNN (N6-Met-X), where freedom of selection in the second amino acid position led to appearance of Pro codon (CCT)

Also, there is inconsistency with the naming.  In the manuscript text and tables, these are primarily referred to as Var #1-6, but in the figures, they are referred to as Koz #1-6.  It is recommended to keep the same nomenclature throughout the manuscript.

We apologize for the inconsistency, it was fixed.

There is no mention in the main text of the data presented in Table 1C.  Additionally, it is not clear what “Inf.Enr.”, “Pack.Enr.” mean, and what the units are for these values.

In order to avoid redundancy, we moved Kozak sequences to Table 1A. On the other hand, packaging and Infectivity can be easy tracked in Table 1S.

In this reviewer’s opinion, Figure S8 is both very interesting and important.  It is recommended to elevate this data to a main figure.  It is also recommended to not stack the transduction values on top of the packaging values, as this masks the values from variant #81 onward.

We elevated the  Figure S8 to the main Figure 4  and eliminated “blockage” of data visibility.

Comments on the Quality of English Language
Minor points:
·       The manuscript requires proof reading to correct grammar errors.  The following are a few examples:
o   Line 74: Please revise to: “Using this approach allowed us to narrow down the most optimized…”
Done
o   Line 99: The motif only shows five Ns (NNNNNATGNN), not six.
Done
o   Lines 109-110: Please revise to: “In this way, Kozak sequences of successfully transduced variants were PCR-amplified and re-cloned into the…”
Done
o   Lines 179-180: Please revise to: “Initially, we intended to identify Kozak variants with optimal transduction efficiencies that rely solely on standard multicycle-directed evolution enhanced Cre-conversion”
Done
o   Line 249: Please revise to: “…we had to use strong GC sequences…
Done
o   Lines 253-261: The authors describe 4 motifs: NNNNATGNATGGC, NNNNGTGNATGGC, NNNNTTGNATGGC, and NNNNTCGNATGGC.  However, the following text in line 261 refers to ATG>GTC>TTG³CTG.  GTC and CTG do not match those described previously.  Please resolve or clarify.
Done
o   Line 452: Please revise to: “This work has resulted in an international patent application: PCT/US2021/029749…”
Done
o   Line 476: Please revise to: “We thank the UF UCBR NGS facility…
Done

Sincerely,

Oleksandr
